# From Traditional to New Benchmark Catalysts for CO_2_ Electroreduction

**DOI:** 10.3390/nano13111723

**Published:** 2023-05-24

**Authors:** Martina Serafini, Federica Mariani, Francesco Basile, Erika Scavetta, Domenica Tonelli

**Affiliations:** 1Department of Industrial Chemistry “Toso Montanari”, Viale del Risorgimento 4, 40136 Bologna, Italy; martina.serafini6@unibo.it (M.S.); f.basile@unibo.it (F.B.); erika.scavetta2@unibo.it (E.S.); domenica.tonelli@unibo.it (D.T.); 2Center for Chemical Catalysis—C3, University of Bologna, Viale del Risorgimento 4, 40136 Bologna, Italy

**Keywords:** CO_2_ conversion, electroreduction, bulk electrodes, gas diffusion electrodes, benchmark electrocatalysts

## Abstract

In the last century, conventional strategies pursued to reduce or convert CO_2_ have shown limitations and, consequently, have been pushing the development of innovative routes. Among them, great efforts have been made in the field of heterogeneous electrochemical CO_2_ conversion, which boasts the use of mild operative conditions, compatibility with renewable energy sources, and high versatility from an industrial point of view. Indeed, since the pioneering studies of Hori and co-workers, a wide range of electrocatalysts have been designed. Starting from the performances achieved using traditional bulk metal electrodes, advanced nanostructured and multi-phase materials are currently being studied with the main goal of overcoming the high overpotentials usually required for the obtainment of reduction products in substantial amounts. This review reports the most relevant examples of metal-based, nanostructured electrocatalysts proposed in the literature during the last 40 years. Moreover, the benchmark materials are identified and the most promising strategies towards the selective conversion to high-added-value chemicals with superior productivities are highlighted.

## 1. Introduction

Carbon dioxide is the second most abundant greenhouse gas in the atmosphere, second only to water vapor [1], and it represents the largest contributor to climate change, although it is not the most powerful one. Indeed, sulphur hexafluoride is considered to be the most dangerous greenhouse gas (GHG), as 1 ton of SF_6_ is equivalent to 23,500 tons of CO_2_, but its current contribution to global warming is estimated to be around 0.2%, due to its very low concentration in the atmosphere, and thus it cannot be included among the compounds that are mainly responsible [2].

In 2013, the Intergovernmental Panel on Climate Change (IPCC) [3] released a report where the importance of carbon dioxide as a greenhouse gas compared to other compounds was demonstrated. By measuring the abundance of GHGs in (i) polar ice cores, (ii) the atmosphere, and (iii) from other climate factors between 1750 and 2011, they achieved a “radiative forcing” value (RF) for each climate driver. Such an indicator, measured in W m^−2^, gives an idea of the influence of each target GHG on the energetic balance between the incoming and outgoing energy from the atmosphere. The RF values for the most responsible GHGs are summarised in Table 1.

Among them, carbon dioxide reported the highest value. Moreover, considering all of the carbon-containing gases, the RF value for CO_2_ increased to 1.82 W m^−2^, confirming its main contribution to climate change, and thus explaining the importance of monitoring its emissions.

The issue of climate change emerged on a global level in the mid-1970s [4]. At this time, the World Meteorological Organization (WMO) started to express some concern about human activities, notably CO_2_ emissions, and in 1979, the First World Climate Conference was held in Geneva [5]. Although it did not attract the attention of many policy makers, scientists of various disciplines agreed about the problem and arranged several working groups with the aim of collecting climate data about its variability and effects on the planet. By increasing the amount of scientific data, and with the discovery in 1985 of the decrease in the stratospheric column density over Antarctica (known today as the ozone hole), reported by Farman [6], human attention on the climate crisis began to rapidly increase.

Although the response of governmental officials to Farman’s article was again initially calm [7], such an event triggered a series of political and scientific conferences, including those in Villach (1985), Hamburg (1987), and Toronto (1988).

On the last occasion, the IPCC [8] suggested identifying measures for the mitigation of GHG emissions and raising awareness via public information and education, setting the stage for climate change negotiations. Hence, although the climate crisis had already been declared at least 20 years earlier, only in the 1990s did it change from a scientific to a political issue, and politicians finally recognised its dangerous potential. Although in the first IPCC assessment report there were no specific national or international targets to limit GHG production, it was a key step for future climate change debates and negotiations [9] that culminated in the conclusion of the Kyoto Protocol in December 1997 [10]. The goal of this international deal, entered into force in 2005 after Russia’s ratification, was to set targets to reduce GHG emissions, in the period of 2008–2012, by at least 5.2%, compared to the levels registered in 1990. To do so, the Kyoto Protocol was based on three market-based “flexible mechanisms” (Figure 1), described in the following scheme:

Moreover, in 2012, a second commitment period was established during the 18th Conferences of the Parties in Doha, known as the “Doha Amendment”. This new phase of the Kyoto Protocol extended the life of such a deal 8 years further (2012–2020). The real change between the first and the second Kyoto period concerned the targets of emission reduction, moving from 5.2% to 18% [12].

Despite several studies indicating that CO_2_ emissions decreased during the Protocol period [13,14], the overall effectiveness of the restrictions was scarce and the main reason was due to the lack of worldwide support, i.e., the United States did not adhere to the protocol obligations and Canada withdrew in 2011 [15].

Indeed, the Kyoto countries emitted about 7% less than those who did not ratify the deal [16], and the developing countries of India and China, who were excluded from the protocol restrictions, contribute nowadays to 27% and 6.6% of global GHG emissions, respectively, together outpacing the United States, set at 11% [17]. Therefore, GHG emissions, and in particular those of carbon dioxide, never stopped growing.

The last main deal written to reinforce the global response to climate change entered into force in November 2016 and it is known as the Paris Agreement. The goals over the long run were set to limit global warming to no more than +2 K, preferably +1.5 K, compared to pre-industrial levels. According to this treaty, starting in 2020, over 140 countries have been committed to achieving net zero CO_2_ emissions, estimating that global GHG emission levels, which were at 52.7 GtCO_2_-eq in 2014, should be cut down to 48 GtCO_2_-eq by 2025, and to 42 GtCO_2_-eq by 2030 [18]. Therefore, over the years, the whole of humanity has become increasingly aware about the climate crisis, trying to limit the anthropological impact. However, the human awareness and responsibility towards climate change have often encountered economic and political obstacles that, on the one hand, have limited the effectiveness of the mitigation policies, while, on the other hand, have pushed the scientific community to put great efforts into developing new strategies in order to guarantee a sustainable future for our society.

To this end, there are a lot of technological solutions that may have the potential to face the problem, including emission reduction, the removal and storage of the already-released CO_2_, and finally its conversion.

Manmade confinement processes can provide higher reductions in carbon dioxide emissions, especially those coming from industries. To date, the focus on carbon capture and utilisation technologies is increasing, as they provide a concrete opportunity to reduce CO_2_ emissions [19]. Indeed, carbon dioxide waste streams can have value as feedstocks for producing more valuable substances which, in turn, are divided into long- and short-lived products [20]. Although the long-lived products, such as cement or durable polymers, are capable of reducing CO_2_ emissions over the long run, the possibility of developing new platforms capable of enhancing the conversion towards fuels and chemicals is attracting major scientific interest. The conversion of carbon dioxide can be achieved via different approaches, which involve thermochemical [21], biochemical [22,23], photochemical [24,25], or electrochemical reactions [26,27]. Their main advantages and disadvantages are summarised in Table 2.

This review focuses on the advantages that electrochemical CO_2_ conversion can afford. Starting from the pioneering work of Hori and his co-workers on the bulk metal electrodes in 1985, a wide range of innovative electrocatalysts have been designed in view of more efficient and cost-effective CO_2_ conversions. Therefore, an in-depth description of metal-based electrocatalyst evolution in terms of shape, functionalisation, productivity, and selectivity on a target product, over the last 40 years, will be provided. Additionally, an insight into benchmark electrodes is emphasised, including the most promising materials to obtain high-value compounds with higher productivities up to 2022. Finally, as for future perspectives, we provide an overview of further optimisations and studies that can be carried out to make the electrochemical approach more appealing from an industrial point of view.

## 2. Electrochemical CO_2_ Conversion

The electrochemical reduction of CO_2_ into C_1_, C_2_, and, sometimes, C_3_ valuable products, such as carbon monoxide (CO), methane (CH_4_), formic acid (HCOOH), methanol (CH_3_OH), ethanol (CH_3_CH_2_OH), ethylene (C_2_H_4_), acetic acid (CH_3_COOH), and propanol (CH_3_CH_2_CH_2_OH) via electron and proton transfers, provides a unique opportunity to develop a green CO_2_ recycle (Figure 2), by virtue of its mild operative conditions, its easily customised reaction outcomes, and its great potential towards scale-up [36,37,38,39].

While in photo-induced CO_2_ reduction the catalyst is directly activated from the photons arising from the sun or a solar-light simulator, electrochemical CO_2_ reduction (CO_2_ER) occurs due to the absorption of artificial electric energy. However, even in the latter approach, the renewable energies may play a key role as the electrons required for the activation of the CO_2_ molecule can be produced from such intermittent alternative energies (solar, wind-powered, geothermal, etc.). Solar-light, for example, can be used either as a direct energy supply for in situ electricity generation (artificial photosynthesis [40]) at the photoanode, or as a way to produce and store electrical energy, exploiting, for instance, solar panels. Both cases ensure eco-friendly alternatives and set the scene towards the fascinating concept of solar-light-driven chemistry [27,40,41]. Moreover, as for photocatalysis, one of the most straightforward strategies for promoting a green CO2 reduction alternative is to carry out the CO2 electrochemical reduction (CO2ER) in aqueous media [42].

However, the solubility of carbon dioxide in water is relatively low compared to organic solvents such as acetonitrile (35 mM vs. 280 mM, at 1 atm and 298 K) [43]; thus, the availability of the reagent decreases if the reaction is conducted in the liquid phase. Another alternative to better dissolve CO_2_ is the use of ionic liquids [44,45], since they provide an environmentally benign medium with respect to the organic solvents and contribute to lowering the energy supply for molecule activation. Despite the beneficial effects of different electrolytes, the use of water as a continuous source of protons is always the best choice in view of a sustainable approach, by virtue of its accessibility, environmental compatibility, and cost-efficiency [46]. However, to seize all of the benefits that the electrochemical approach can provide and to make it affordable from an industrial point of view [47], many challenges still need to be solved.

Unlike the thermochemical approach, electrochemical systems offer the possibility to work under ambient pressure and temperature, but high overpotentials are usually required for the activation of the carbon dioxide molecule [48]. Generally, such a reduction reaction suffers from sluggish kinetics, multi-phase rate limiting steps, and poor selectivity, as well as intermediate-sharing or multiple reaction pathways, whose mechanisms are still under debate. Moreover, experimental parameters such as catalyst properties, local pH, chemical nature of the electrolyte solution, and the applied potential dramatically affect the reaction and, consequently, the product distribution [34,49,50].

When performing the CO2 electrochemical reduction in aqueous media, the hydrogen evolution reaction (HER) inevitably occurs due to the similar potentials. For the sake of clarity, the standard redox potentials of HER and of the first two-electron transfer products that can be obtained from the CO_2_ER reaction are reported below [51]:2H^+^ + 2e^−^ ⟶ H_2_    E^0^ = −0.41 V(1)
CO_2_ + 2H^+^ + 2e^−^ ⟶ HCOOH   E^0^ = −0.61 V(2)
CO_2_ + 2H^+^ + 2e^−^ ⟶ CO + H_2_O   E^0^ = −0.53 V(3)

HER is more kinetically favoured than the multi-electron transfer reactions of the CO_2_ reduction [52], thus greatly affecting the proton availability and, consequently, the pH at the electrode–electrolyte interface. Therefore, a lot of efforts have been put towards the study of novel electroactive materials capable of enhancing CO_2_ conversion, thus improving the control over the selectivity and limiting the HER in aqueous media. Overall, the understanding of the reaction mechanisms that affect the electrocatalytic performances, alongside the changes in the catalyst structure, is of particular interest. To this end, in situ and operando techniques of a spectroscopic or microscopic nature have gained great interest as promising approaches for investigating the electrocatalyst, along with the binding modes of the intermediates, during CO_2_ER reactions [53,54]. In particular, operando and in situ X-ray absorption (XAS) [55], X-ray photoelectron spectroscopy (XPS) [56], infrared [57], and Raman [58] spectroscopies are considered to be the most promising techniques. Additionally, the combination of such advanced characterisation techniques (particularly, in situ surface-enhanced Raman spectroscopy (SERS), in situ electrochemical transmission electron microscopy (TEM), and in situ XPS) and density functional theory (DFT) [59] calculations is currently gaining momentum towards the disclosure of reaction mechanisms and involved active sites, as has been well summarised in a recent review by Wang et al. [60].

Today, the scientific community generally acknowledges that the CO_2_ER reaction likely follows two main pathways, i.e., the first preferentially leading to the formation of C_1_ products and the second one preferentially leading towards the formation of C_2/2+_ (products with two or more carbon atoms) products. Figure 3 shows a scheme of the two possible abovementioned mechanisms occurring on the Cu catalyst surface.

In light of the large variety of compounds that can be obtained, many efforts are being made in order to tune the selectivity of the employed material towards the formation of the desired product, investigating the effects of the reaction environment on the product distribution, such as the catalytic layer, local pH, or the applied potential [62,63].

As far as the electrolytes are taken into account, the most widely employed one to test the activity of a target electrocatalyst in aqueous phase CO_2_ER is KHCO_3_ [64,65,66,67]. Zhong et al. [68] carried out studies on the effect of CO_2_ bubbling inside different electrolytes, i.e., KHCO_3_, K_2_CO_3_, KOH, KCl, and HCl. Along with theoretical calculations, the authors demonstrated that the best electrolytes to perform the CO_2_ER reaction were KHCO_3_ and KCl because they provided a high amount of active carbonate species, i.e., H_2_CO_3_ or HCO_3_^−^, which were assumed to favour a higher efficiency. Moreover, the pH values after 10 min of CO_2_ bubbling were stable, being neutral or slightly alkaline for KHCO_3_ and acidic for KCl, while avoiding strong pH variations up to 4–5 units as registered with KOH or K_2_CO_3._

More recently, Koper’s group [69] investigated the importance of acid–base equilibria in hydrogen carbonate solutions when employing Au electrodes, and found that the fundamental role of the HCO_3_^−^ species was acting as a further source of CO_2_. They also carried out several studies on the nature of the cation present in the electrolyte, which likely plays an important role in terms of reaction selectivity [70]. They demonstrated that the formation of the hydrogenated dimer (OCCOH), which represents the initial intermediate for the formation of C_2_ products, was related to the size of the cation. Indeed, it was preferentially formed at low overpotential in the presence of smaller ions such as Li^+^, K^+^, and Na^+^, thus evidencing that a specific cation can stabilise a desired intermediate. Moreover, they found out that a metal cation, regardless of its nature, is, in all cases, fundamental in activating the reduction process as it forms a complex with CO_2_ that promotes the formation of the CO_2_^–^ intermediate [71]. Indeed, they observed no CO_2_ reduction products in silver, copper, and gold electrodes without any metal cations in the electrolyte. As far as the reaction conditions at the electrolyte–electrode interface are concerned, they have recently carried out investigations about the interfacial pH, using rotating Au ring-disk electrodes, as a function of different electrolytes [72], revealing that the pH gradient between the electrode and the bulk electrolyte plays a fundamental role in CO_2_ER efficiency.

Furthermore, Schuhmann’s group has reported several studies concerning the optimisation of the gas–liquid–catalyst interphase. In a recent work, B-doped Cu nanoparticles were designed with the aim of stabilising the active Cu^+^ species in the catalyst, and the HER contribution consistently decreased via the fine tuning of the gas diffusion electrode (GDE) hydrophilicity by adjusting the PTFE loading in the catalyst. Moreover, its structural stability against cathodic corrosion was improved by introducing Zn as a sacrificial species. A Faradaic efficiency (FE) of 78% for the formation of C_2+_ products was achieved using a 0.5 mg cm^−2^ B-Cu GDE with 10% PTFE, at a potential of −0.45 V vs. RHE, reaching a remarkable current density of −200 mA cm^−2^ [73]. Furthermore, scanning electrochemical microscopy (SECM) was exploited to investigate the local pH changes [74] on the surface of the nanostructured Cu_x_O_y_C_z_/GDE electrocatalysts upon PTFE loading variation. A similar approach was then employed, in collaboration with Koper’s group, to monitor the locally generated CO under operando conditions [75] using a Au/C nanoparticle-sprayed GDE, as a function of catalyst loading and CO_2_ back pressure.

All of the aforementioned studies have been carried out in heterogeneous configurations, which represent the most promising electrocatalytic routes. However, it is worth noting that electrochemical CO_2_ reduction can be carried out using both homogeneous and heterogeneous electrocatalysis, although commercial systems are not available today. However, between the methods, the less suitable for industrial applications is represented by the homogeneous one, which has nevertheless been widely studied by the scientific community for decades [76].

With regard to the conversion of carbon dioxide, such an approach aims to facilitate redox mediation in the liquid phase, taking advantage of a homogeneous catalyst dispersed into the electrolytic bath acting as a redox shuttle between the electrode and CO_2_. Briefly, once the catalyst has been reduced, it interacts with the carbon dioxide present in the solution and carries out the relevant redox reduction. The most widely investigated homogeneous electrocatalysts are the metal–organic compounds designed to mimic the biochemical CO_2_ conversion and capable of operating close to the thermodynamic potential of the reaction, if properly tuned with specific ligands [77,78]. Moreover, by virtue of their well-defined and controlled molecular structure, the optimisation of the selectivity is much easier, but there are several drawbacks that significantly reduce their applicability in industry, with the most important challenging ones being the recovery of the catalyst and the separation of the products [76].

Despite a lot of studies concerning homogeneous CO_2_ electroreduction [76,79,80], the heterogeneous approach still appears to be the most suitable for industrial applications and a more in-depth discussion will be provided. As the name suggests, this kind of electrocatalysis concerns reduction reactions that occur at the electrode–electrolyte interface [81]. Unlike the homogeneously catalysed reaction, the catalyst is directly attached on the surface of the electrode, forming a single system acting both as the acceptor and donor of electrons, which can be exploited for reactions in both the liquid and gaseous phase.

Theoretically, the first step of the reaction is the chemical adsorption of the carbon dioxide on the surface of the electrocatalyst, followed by the formation of the intermediates via the cleavage of the C=O bond, carried out by electrons and protons. Once the intermediate is rearranged into the target product, it is desorbed from the electrode surface and dispersed into the bulk electrolyte [82]. In a typical H-cell for CO_2_ electroreduction, the cathode and the anode are placed in two different chambers separated with an ion-conducting membrane with the aim of avoiding the further oxidation of the reduced products of the reaction. Indeed, on the anode side, water is oxidised to molecular oxygen and protons are produced, while CO_2_ is reduced to different carbonaceous species on the cathode side [27]. The choice of the membrane depends essentially on the pH of the electrolyte, but the mainly employed membranes are made of Nafion as they favour H^+^ diffusion from the anode to the cathode side. The majority of the well-known applications of electrocatalysis such as fuel cells, chlor-alkali electrolysers, and the water splitting process are examples of heterogeneous electrocatalysis. Schemes of he most representative cell designs employed for the CO2ER reaction are reported in Figure 4.

The rate of all of these reactions, and therefore also of the CO_2_ER reaction, as mentioned above, depends on the electrochemical potential gradient at the electrode–electrolyte interface [84], the activity of the electrocatalyst, and the type of electrolyte. Since water has been recognised as the most useful source of protons to be implemented inside the electrocatalytic cells in view of sustainable and cost-effective systems, the type of electrocatalyst naturally becomes the key challenge.

Prior reports have reported an excellent overview of possible products that can be obtained from electrochemical CO_2_ reduction, including the following: CO, formate, methane, ethylene, ethanol, n-propanol, allyl alcohol, acetaldehyde, propionaldehyde, acetate, methanol, ethylene glycol, glycolaldehyde, hydroxyacetone, acetone, and glyoxal [85,86]. In the CO_2_ER scenario, there are some important indicators that need to be considered in order to classify the performances of the overall reaction set-up, and thus its industrial applicability [87]. Since the process costs have not yet been scheduled into standard protocols, the most common provided parameters are Faradaic efficiency and current density (J). Another important value that aims to give an indication of the energy required to overcome the energetic barrier for the activation of the CO_2_ molecule is the overpotential (*η*). Given as an absolute value, this term symbolises the difference between the equilibrium potential for the relevant CO_2_ER half reaction and the applied potential at which a target product is produced. The required overpotential is often an indicator of the efficacy of a catalyst to promote the reaction: the less overpotential is needed, the more active the electrocatalyst is.

Faradaic efficiency represents the selectivity of the overall reaction towards a given product, in terms of the percentage of electrons that are transferred to the products. High selectivity, and thus a high FE, for the target product is advantageous to avoid wasting energy in the production of unwanted compounds, and also to simplify product separation. It can be expressed as the ratio between the theoretical charge needed for the production of a defined amount of product and the overall charge passed during the reaction, according to the following equation:(4)$$\mathrm{FE}\;\left(\%\right)=\frac{n\;\times \;z\times F}{Q}\;\cdot 100$$
where n are the number of moles obtained for a target product, z is the number of electrons exchanged during the reaction to obtain the product, F is Faraday’s constant (*F* = 96,485 C mol**^−^**^1^), and Q represents the overall charge.

The other important performance indicator is the current density (J), which is commonly used in heterogeneous electrocatalysis to define the average activity of the catalyst since active sites with different structures often coexist. It defines the rate of the electrochemical reaction [81], as the turnover frequency does in the homogeneous electrocatalysis [88], and it is calculated by dividing the current developed as a result of a given cell potential by the geometrical or the electrochemical surface area of the electrode. Since the latter value is not always easy to determine, the geometrical area is the most used as it also gives an indirect indication of the size of the electrochemical cell, and thus of the overall necessary investment.

The following discussion will be focused on the evolution of the materials employed as heterogeneous electrocatalysts for the CO_2_ conversion, performed in aqueous media or in electrolyte-less systems that exploit water as s source of protons on the anode side. Henceforth, the heterogeneous CO_2_ electroreduction will be referred to as the CO_2_ electroreduction.

### 2.1. Traditional Electrocatalysts

The first studies on electrochemical CO_2_ reduction date back to the 1950s, when Teeter et al. [89] reported the formation of formic acid using a Hg-based cathode. However, the most relevant studies that include a total quantification of both gaseous and liquid products were performed by Hori and co-workers [90,91,92,93] and were mainly based on monometal bulk catalysts, especially the transition ones. Starting from this research, they were divided into four categories, depending on the primary CO_2_ reduction product [34,94]: (i) CO, (ii) HCOOH, (iii) H_2_-selective materials, and (iv) beyond CO. Such a classification was established after an exhaustive study carried out by Hori et al. [64] in 2008, in which he showed different performances of each bulk metal electrode during the galvanostatic CO_2_ electroreduction. In these experiments, the current density was kept constant at 5.0 mA cm**^−^**^2^, along with the pH that was fixed at 6.8, using a 0.1 M KHCO_3_ electrolyte, and stabilised under operating conditions.

The metal electrodes capable of producing CO include noble metals such as Au, Ag, Pt, and Pd [95], but also non-noble metals such as Cu, Zn, Ga, and Ni [86].

Among them, Au and Ag provide the highest catalytic activity with a maximum FE of 87% for Au. However, due to their low abundance on Earth and high cost, up until now, they have not been considered for large-scale industrial applications [96]. Hence, since Zn has demonstrated similar performances, it represents the most promising metal electrode for the selective production of CO. A very recent review by Wu et al. summarises the state of the art concerning Zn-based catalysts, including Zn monomers, Zn-containing bimetals, oxide-derived Zn catalysts, and single/dual Zn atom catalysts for the CO_2_ER reaction [97].

The second class of bulk metal electrodes is represented by Pb, Hg, Tl, In, Sn, Cd, and Bi, which produce HCOOH as the primary product. A noticeable FE in formic acid production was achieved with Pb and Hg (up to 99.5%) [86]. Differently from the CO, the production of formate/formic acid (depending on the operating pH) involves the use of protons and thus it follows a different pathway. Despite the mechanism still being under debate, the most accredited route includes the formation of the *OCHO intermediate [82]. To date, Sn and Bi have attracted a lot of interest for their large-scale use because of their relevant catalytic activity, environmentally friendly nature, and economical convenience [98].

The third class of monometal electrodes includes materials with very low activity for CO_2_ electroreduction, but with substantial efficiency for the hydrogen evolution reaction. Indeed, bulk electrodes such as Pt, Ni, Fe, and Ti displayed FE values that reached up to 90%.

Finally, concerning the last group, Cu has been overall recognised as the gold standard among pure metal catalysts [34,90,99] due to its unique ability to catalyse CO_2_ER beyond CO, i.e., towards a number of hydrocarbons, aldehydes, and alcohols requiring more than two-electron transfers with high FE. In particular, the main advantage of Cu electrodes relies on their capability to stabilise both the chemisorbed CO_2_^−^ radical anions and CO species, which are the key intermediates in the initial phase of the catalytic CO_2_ conversion and in the subsequent process of hydrocarbon and alcohol formation, respectively. In its bulk form, it was the only material capable of producing methane, ethylene, ethanol, and propanol, along with low amounts of carbon monoxide and formic acid [64]. For this reason, starting from those pioneering studies, nowadays, copper is considered to be the most promising material to be used in electrochemical CO_2_ reduction.

Along with the polycrystalline bulk metals, single-crystal metals have also been widely studied, finding out that a specific crystalline lattice rather than others can considerably enhance the overall production or tune the selectivity [100,101,102,103].

Moreover, it is worth mentioning that among the traditionally investigated electrocatalysts, the properties of the metal oxide materials are already well-known in the heterogeneous catalysis field and these compounds have also been widely used in the CO_2_ER reaction, bringing significant improvement in the overall performance, when compared with metal electrodes [104,105,106,107].

As an example, Deng et al. [108] showed that Bi-O systems can lower the energy barriers for intermediate formation in the HCOOH production pathway, thus demonstrating the importance of oxygen species in switching the rate-determining step (Figure 5).

Recently, Gao et al. [109] verified how the presence of the metal oxide interface during the CO_2_ER reaction can considerably enhance the gas adsorption on the surface of the catalyst. However, studies achieved by means of advanced in situ characterisation techniques have reported the instability of the metal oxide species under CO_2_ electroreduction conditions [110,111]. Such an instability not only affects the oxidation state of the metal active phase but also its structure. For instance, one of the main issues concerns the possible role of surface copper oxide as an active catalyst towards C_2+_ products, whereby there is still an open debate concerning this with conflicting opinions. On the one hand, several researchers have revealed that Cu^+^ species survived even after the beginning of the CO_2_ reduction, also reporting that the residual sub-surface oxygen changes the structure of the oxide-derived Cu electrocatalyst, thus shifting the selectivity towards C_2_ products [112,113]. On the other hand, Ren et al. [114] have proven via in situ Raman analyses that, upon the application of cathodic potential, the Cu_2_O-based electrocatalyst was fully reduced to Cu^0^ within a few minutes, thus suggesting metal copper as being the real active phase. Moreover, additional in situ Raman analyses have demonstrated that restructuring processes upon cycling from oxidising to reducing potentials were useful for enhancing the selectivity towards C_2_ products [115]. However, thin oxide layers over the bulk electrode generally promote the formation of nanostructured interfaces, which will be discussed in the following paragraph. For these reasons, the origin of the higher catalytic activity of the metal oxide species compared to the bulk metal electrodes may result from various aspects, and thus it is still under debate [116].

Therefore, great efforts are being carried out on the design of the electrocatalysts in order to achieve competitive performances and higher energetic efficiencies. To this end, a new generation of electrocatalysts is taking hold for the CO_2_ER reaction with advanced structures and new reactive sites that display attractive qualities over bare bulk electrodes. Different materials in terms of size of the catalyst particles, oxidation state, porous hierarchical structure, doping, alloying, and defect engineering have been developed as novel selective and active electrocatalysts with the aim of outpacing the limitations of the CO_2_ER reaction, in view of feasible and greener processes for future CO_2_ electrolysers. Among them, nanostructured materials, nano- or microporous films, oxide-derived metals, and carbon-based materials are some examples of the more recent advanced systems that, to date, represent the leading electrocatalysts for electrochemical CO_2_ reduction in aqueous media.

### 2.2. Electrocatalysts of the New Generation

The major parameters that significantly impact the final outcome of the CO_2_ER reaction are the electronic and geometrical structure and the composition of the materials [81]. In particular, the required overpotential and the selectivity are mainly affected by the changes in the electronic structure, while the geometric structure is related to the catalytic sites’ density, and thus influences the current density recorded during the reaction.

The size and morphology of the active catalyst play very important roles in the overall CO_2_ER efficiency. In particular, it has been widely demonstrated that reducing the dimensions from macro- to nanosized materials enhances the number of superficial active sites, and thus the available electrochemical surface area of the catalyst [117]. To this end, various methods have been developed including bottom-up or top-down approaches [118], and the principal characteristics that discriminate a nanostructured or nano-electrocatalyst from a bulk one are the increase in the catalyst surface per mass and the introduction of many edge/low-coordinated sites [51,119]. Notably, several studies have been carried out on nanosized metal catalysts, including metal nanoparticles (metal NPs) [120,121,122,123], nanocubes [115,124], and nanostructured catalysts [27,125].

Creating nanostructures over the surface of a bulk electrode can substantially outpace the reaction performances of its bare polycrystalline counterpart. As examples, Zhu et al. [126], Liu et al. [127], and Zhu et al. [128] demonstrated that Au nanowires, Ag triangular nanoplates, and ultrathin Pd nanosheets, respectively, increased the surface ratio of their edge sites more than their spherical nanoparticle counterparts, thus balancing the *COOH/*CO adsorption ratio with a beneficial effect on the overall reaction performance. Additionally, an early study carried out by Li et al. [129] reported outstanding efficiency and selectivity towards the production of formate using a 3D hierarchical structure composed of mesoporous SnO_2_ nanosheets with an average pore diameter of 4–5 nm. The active phase, supported on carbon cloth, exhibited an unprecedented current density of 50 mA cm**^−^**^2^ with a total FE of about 87%. Despite the reaction performance being comparable to the one achieved using the bulk Sn electrode, whose FE for formate was 88% [64], the possibility to obtain similar results using a less-loaded catalyst (0.34 mg cm**^−^**^2^) and a flexible and sustainable support was demonstrated, thus avoiding heavy bulk electrodes.

Moreover, inside the group of nanostructured and porous materials, catalysts such as layered double hydroxides (LDHs) or metal–organic frameworks (MOFs), which are usually employed for the preparation of photocatalytic reduction devices [130], are gaining increasing interest as they are also suitable for electroreduction applications.

Among all of the transition metals, the majority of the studies performed on the morphology of the catalysts have found it using Cu, which was found to be facet/shape-dependent [59,131,132]. Although the description of the underlying reaction mechanisms is still challenging, several researchers have demonstrated the possibility of tuning the reaction selectivity by manipulating the copper morphology and size. A comparative study among Bismuth−based catalysts for the CO2 electrochemical reduction to formate, Electropolished, and Sputtered with Cu NPs was performed by Tang et al. [133], in which better selectivity for hydrocarbons was evidenced for the first kind of electrode. The reactions were carried out in aqueous media, using KClO_4_ as inorganic salt, and applying a potential of −1.1 V vs. the reversible hydrogen electrode (RHE). In particular, the investigation showed the outstanding performances of the nanoparticles’ covered surface which displayed the highest C_2_H_4_/CH_4_ ratio. Such evidence clearly highlights the importance of the morphology of the employed copper, beyond the reaction conditions. Moreover, stressing this feature, Suen et al. [134] recently reported the tremendous shift in the reaction selectivity from C_2_ to C_1_ products via changing the morphology of copper from cube-like and hexarhombic–docadehedron-like Cu single crystals to octahedron-like Cu nano single crystals (Figure 6).

However, several changes may sometimes cause undesired effects on the product selectivity towards hydrocarbons. Reske et al. [135] were some of the first to evaluate the effect of different Cu nanoparticle sizes (2–15 nm), with respect to the bulk electrode. They demonstrated that particles with a dimension below 5 nm exhibited a dramatic decrease in the catalytic activity towards hydrocarbons, thus favouring the production of H_2_ and CO. Moreover, although a higher selectivity for hydrocarbons was observed for the 5–15 nm particles, the FE in CH_4_ and C_2_H_4_ dropped in respect to their bulk counterpart (10–15% vs. almost 60% for CH_4_, and 0–15% vs. 20% for C_2_H_4_). Hence, if the purpose is hydrocarbon production, the proper Cu nanoparticle dimension must be considered. Indeed, due to the well-known changes in the material properties on the nanoscale dimension [136], other transition metals displayed size effects along with copper. On the one hand, by decreasing the Pb nanoparticles’ size (2.4–3.7 nm), an increase in CO production was observed [120], while, on the other hand, a maximum CO FE was obtained employing middle-size Au nanoparticles (8 nm) [137].

Furthermore, an alternative method for enhancing the density of the active sites consists of an in situ reduction of the catalyst, starting from its oxidised form, whereby the final catalyst is referred to as an oxide-derived electrocatalyst. Such a phenomenon is carried out via oxidation and reduction processes that lead to the formation of defects and roughened surfaces. As an example, in the case of Cu, it has been demonstrated that using oxide-derived copper species enhances the selectivity towards C_2_ products [50].

Along with the morphology and size of the catalyst, another important parameter that contributes to selectively changing the reaction outcomes is the oxidation state of the active phase. Although higher oxidation states are unstable under the reductive reaction conditions, several pieces of experimental evidence have confirmed the increase in catalytic activity by modulating the oxidative state of a material. As an example, the addition of S to a Sn catalyst allowed it to reach a current density of 55 mA cm**^−^**^2^, and there was a total FE for HCOOH of 93%. In this case, the electronic properties of Sn significantly changed due to the fact that the oxidation state was between 0 and +2 [138].

Concerning the catalytic activity of copper, there are still many open questions regarding the stability of its oxidised forms during the CO_2_ER reaction and the possible role that they may have in the reaction pathways. However, several pieces of experimental evidence have highlighted the fundamental role of the Cu^+^ species towards the production of C_2_ products [139].

Apart from the bulk metal electrodes that are generally employed as such, the new generations of catalysts, especially nanostructured or porous materials and metal-based nanoparticles, need to be loaded onto a conductive support. Such systems can be considered as composite materials made either from different forms of the same materials (e.g., bulk electrodes covered by a metal nanostructured layer) or from different materials (e.g., metal NPs supported on carbon fibres). Since the electrochemical CO_2_ER reaction has been usually evaluated employing a traditional H-type cell [140] thanks to its simple configuration, the geometry and the nature of the conductive support, along with the nature of the active phase, are important features to be considered. The main problem that occurs when the reaction is performed in an H-type cell is the mass transfer limitation due to the low CO_2_ solubility, which directly limits the current density usually below 30 mA cm**^−^**^2^ [141]. The catalyst active phase can be supported either on planar or non-planar electrodes, such as porous/3D materials. However, the use of planar electrodes contributes to lowering the current density of the reaction due to the slow CO_2_ diffusion towards the surface of the conductive material [142,143].

To overcome these limitations, porous/3D porous materials have been preferentially employed as they can act both as active phase support and as an interface between the gaseous reactant and the catalyst layer. The most commonly used materials are metal foams [144,145] or carbonaceous gas diffusion layers, such as carbon fibres or carbon nanotubes [146,147].

Porous/3D electrodes are greatly desirable in energy conversion applications, since they not only possess large surface areas to increase the number of active sites and provide a high degree of dispersion of the catalyst, but also decrease the contact resistance and, hence, facilitate the electron transfer. Therefore, interconnecting macro- or microporous structures with nanostructured catalysts likely brings about an optimal set-up to overcome the current density challenge and reduce the energy barriers for CO_2_ activation (overpotential—η). Such systems, designed to interact with a gaseous phase, can also be employed in the CO_2_ER liquid phase as their hydrophobic nature might trap the CO_2_ near the catalyst layer to locally form solid–liquid–gas interfaces, which could improve the activity and selectivity for CO_2_ reduction [140]. As for the photocatalytic applications, carbonaceous materials, especially large-area gas diffusion layers, are gaining increasing interest for their electrochemical applications as well. Indeed, thanks to their large availability and eco-friendly nature, they embody the most sustainable alternatives for the preparation of cost-effective electrodes with promising performances regarding the CO_2_ER reaction, and easy scalability. As an example, one of the latest devices which involves the use of carbonaceous gas diffusion layers and allows one to further improve the reaction efficiency is the gas diffusion electrode configuration [148]. It facilitates the diffusion of the gaseous reactant towards the catalytic layers as the electrode is located between a continuous gaseous flow (the electrolyte-less compartment) and the supporting electrolyte connected to the anode side.

### 2.3. Benchmark Electrocatalysts

Since entering the 21st century, the number of the investigated catalysts and the related research about the CO_2_ER reaction have exponentially increased [82]. Nowadays, the principally investigated catalysts are nanostructured/porous materials, composed of monometal or metal alloys at different oxidation states, depending on the desired product. The catalytic active phase is generally loaded onto 3D conductive electrodes, mainly carbonaceous, which offer the possibility of being employed for the CO_2_ER reaction in both the liquid and gaseous phases.

Although the applicability of these systems is still under the standard levels for industrial applications that require very high current densities (200–300 mA cm^−2^) alongside a selectivity of 90% and a CO_2_ conversion of 50% [149], big efforts have been made from a scientific point of view, as already stated previously.

This paragraph aims to give a brief overview of the leading catalysts that have displayed outstanding performances in aqueous media for CO_2_ conversion towards the desired product. Wu et al. [150] reported an exhaustive list of the main electrocatalysts, which illustrates the recent strategies used to enhance the FE with respect to the target products.

Concerning the two-electron transfer products, in 2018, Liu et al. [151] carried out a CO_2_ER reaction with an aim of improving the conversion towards the production of CO by means of 5-fold twinned Ag nanowires supported on a glassy carbon electrode, with diameters less than (i) 25 nm (D-25) and (ii) 100 nm (D-100). As a result, the D-25 Ag NWs displayed the highest conversion to CO, up to 99.3%, and a current density of 2.2 mA cm^−2^ upon the application of −0.956 V vs. RHE in 0.1 M KHCO_3_. Additionally, an outstanding conversion from CO_2_ to formate was reported by Zhang et al. [152], who were able to obtain an FE for HCOOH of ~91% at −0.9 V vs. RHE using pipet-like N-doped carbon nanotubes semi-filled with Bi nanorods (Bi-NRs@NCNTs) supported on a GCE. The reaction was performed in an H-type cell, employing 0.1 M KHCO_3_ as a supporting electrolyte, under ambient conditions. Recently, three-dimensional Bi_2_O_3_ nanofoams supported on a carbon fibre cloth were reported for the CO_2_ER to formate conversion, with evidence of two different reaction pathways in the aqueous-CO_2_-saturated 0.5 M KHCO_3_, i.e., (i) a sub-carbonate pathway in the partly reduced Bi_2_O_3_ foam with low overpotentials (reaching 97.3% FE at −0.6 V vs. RHE) and (ii) a Bi–O pathway in the corresponding metallic Bi foam catalyst with medium and high overpotentials (reaching 91.7% FE at −1.0 V vs. RHE) [153]. Similarly, Feng et al. designed a Bi_2_O_3_/BiO_2_ heterojunction catalyst working in 0.5 M KHCO_3_ inside a microfluidic flow cell electrolyser and reaching >95% FE for formate within a wide potential range from −0.9 V to −1.3 V vs. RHE [154]. A carbon-confined In_2_O_3_ catalyst was prepared via high-temperature calcination, under an Ar atmosphere, of the precursor indium–organic framework (MIL-68(In)) [155]. When tested in a liquid phase flow cell, In_2_O_3_@C reached high selectivity (ca. 94%) for formate production from −0.8 V to −1.3 V vs. RHE, outperforming both the bare In_2_O_3_ and bulk-In_2_O_3_ catalysts. In another recent report, the facet-dependent activity of Cu_2_O/hollow fibre catalysts was investigated for the conversion of CO_2_ER to formate [156]. In particular, the (111)-oriented Cu_2_O-based catalyst, thanks to the synergy with abundant oxygen vacancies, reached a 92.3% FE (−1.18 V vs. RHE) with a partial current density of 84.4 mA cm^−2^ during a 1 h reaction in CO_2_-saturated 0.5 M KHCO_3_.

Concerning the multi-electron transfer products, the most common compounds reported in the literature are CH_3_OH, CH_4_, C_2_H_4_, CH_3_CH_2_OH, CH_3_COOH, and CH_3_CH_2_CH_2_OH (n-propanol). Except for methane, whose higher FE (85%) was obtained by exploiting Zn atoms supported on microporous N-doped carbon (SA-Zn/MNC) [157], all of the catalysts employed to carry out the reaction towards >2e^−^ transfer products included the presence of copper.

Indeed, the highest FEs for the three alcoholic species were obtained using three Cu-based catalysts of different morphologies, according to the previously mentioned morphology-dependent selectivity towards C_2_ or C_1_ products. In particular, CH_3_OH was produced with a copper selenide nanocatalyst supported over 1 cm^2^ sized carbon paper (Cu_1.63_Se(1/3)), with an FE of 77.6% [158] (Figure 7), while a carbon-supported Cu catalyst—Cu_n_ (*n* = 3 and 4)—cluster was used to obtain CH_3_CH_2_OH with an FE of 91% [159]. On the other hand, n-propanol, which requires both the stabilisation of *C_2_ intermediates and subsequent C_1_–C_2_ coupling, was preferentially achieved on a hexagonal double-sulphur vacancy-rich CuS catalyst (CuS_x_—DSV) with an FE of 15.4% [160]. Differently, as for the production of acetate, until 2017, the highest FE (~72%) was reported by Marepally et al. [161], employing Cu^0^ NPs on carbon nanotubes (Cu^0^ NPs@CNTs) upon the application of −1.35 V vs. RHE. However, our group recently developed new Cu-based electrocatalysts with enhanced acetic acid productivity that outpaced the catalytic performances already described in the state of the art. In particular, by exploiting electrosynthesised CuMgAl LDHs, which are biocompatible inorganic lamellar nanomaterials [162,163], on a carbonaceous gas diffusion layer (CuMgAl LDH/CP), the selective production of acetic acid at −0.4 V vs. RHE was successfully obtained, with an FE of 84% [164].

Finally, a comparison between the Faradaic efficiencies of the most relevant reduced products obtained using the benchmark catalysts described above and the results achieved via Hori’s pioneering studies on bulk metal electrodes is reported in Table 3.

## 3. Conclusions and Future Perspectives

Electrochemical methods are gaining momentum in the management of the energy transition that we are experiencing, including the pursuing of competitive carbon capture and utilisation strategies. In this review, we have tried to describe the evolution of the metal-based electrocatalytic systems that have been reported so far for the conversion of CO_2_ into fuels, eventually depicting the complex roadmap of the challenges, targets, and achievements concerning such a demanding technological application.

The new-generation electrocatalysts, characterised by the presence of nanomaterials and nanostructures, have led to a substantial outpace in the reaction performances achieved by the bare polycrystalline counterparts. The reason for such a dramatic improvement must be sought in the higher density of catalytic sites and the increased electrochemical surface area of the catalyst, thus leading to a larger current density recorded during the reaction. Along with the establishment of the green chemistry criteria for the scientific and industrial research, and together with the urgent need for alternative and less-impacting energy sources, the interest in the electrochemical reduction of CO_2_ has been rising sharply in the 21st century. The currently most promising electrocatalysts rely on nanostructured materials including one or more active metals and alloys, with tuned and controlled morphologies and electronic structures, supported on three-dimensional conducting gas diffusion electrodes, allowing the creation of a triple phase boundary.

Overall, great advancements have been made from a material design point of view in order to cope with the high overpotentials, sluggish reaction kinetics, and multiple reaction pathways characterising CO_2_ER. However, with this research field being highly technologically driven, there is a strong push for overtaking the proof-of-concept stage towards the development of systems that show compatibility with applicative scenarios and show promise in meeting industrial standards. Nevertheless, there are inescapable gaps to fill before a competitive, scalable, and sufficiently efficient electrochemical conversion of carbon dioxide can be realised in terms of current densities, selectivity, and conversion. The following gaps are highlighted in particular:

(i)Even if a growing number of reports concerning the local investigation of the electrode/electrolyte interface are being released using in situ and operando techniques, the thorough understanding of reaction mechanisms has not been fulfilled yet. Improved knowledge about local phenomena occurring on the nano- and microscale would be greatly beneficial for the optimisation of the catalyst composition and morphological properties depending on the experimental conditions of choice. This should impact the overall, macroscopic, electrochemical performances, for instance, lowering the required overpotential, enhancing the productivity towards the desired species, and improving the catalyst stability and regeneration capability. To this end, the interplay among different interfacial techniques as well as computational modelling is highly encouraged.(ii)CO_2_ER systems are often investigated regardless of the anodic part of the electrochemical cell. Not only should the anode not be ignored due to its impact on the overall cost and sustainability of the final device, but it could also be exploited to combine the conversion of CO_2_ with another valuable electrochemical reaction, such as oxygen evolution or wastewater treatment. From this point of view, the design of versatile, multi-functional electrocatalysts would represent a further improvement in the overall efficiency of the cell.(iii)Different from other electrochemical processes for energy storage and production, no standardised protocol or benchmark has been established for the electrochemical reduction of CO_2_ yet [165]. The development of such tools would help the objective evaluation and comparison of electrocatalysts’ performances.

We strongly believe that these aspects, together with the ongoing material research, will play a key role in the next future evolution of CO_2_ER systems and their effective contribution in managing energy transition.

## Figures and Tables

**Figure 1 nanomaterials-13-01723-f001:**
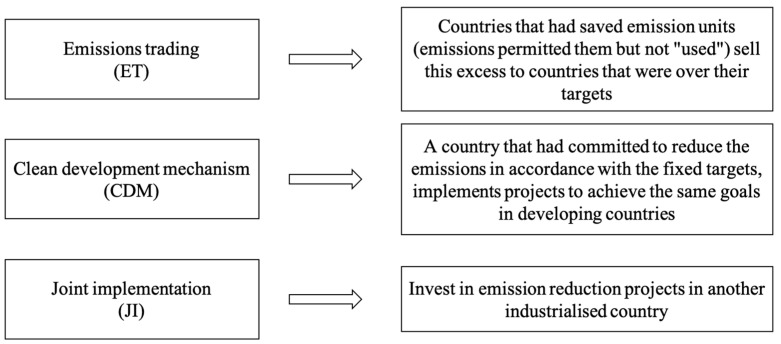
The Kyoto Protocol market-based mechanisms [11].

**Figure 2 nanomaterials-13-01723-f002:**
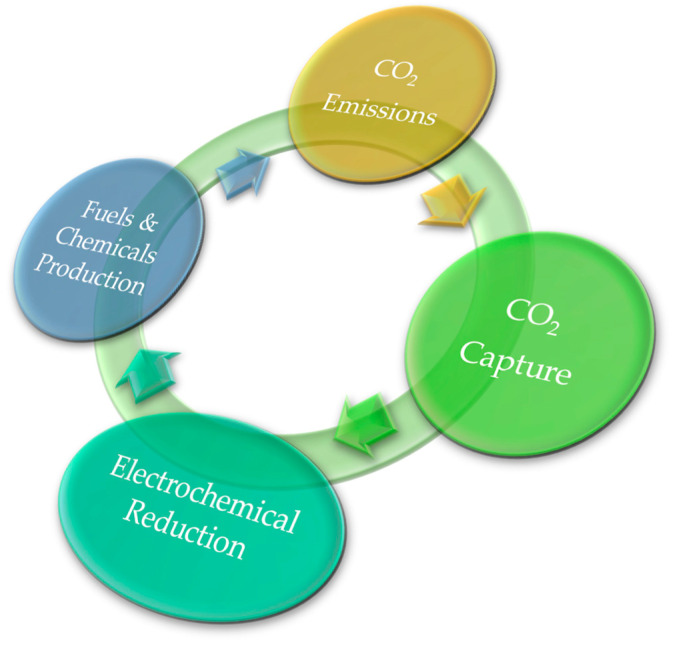
Green CO_2_ recycle.

**Figure 3 nanomaterials-13-01723-f003:**
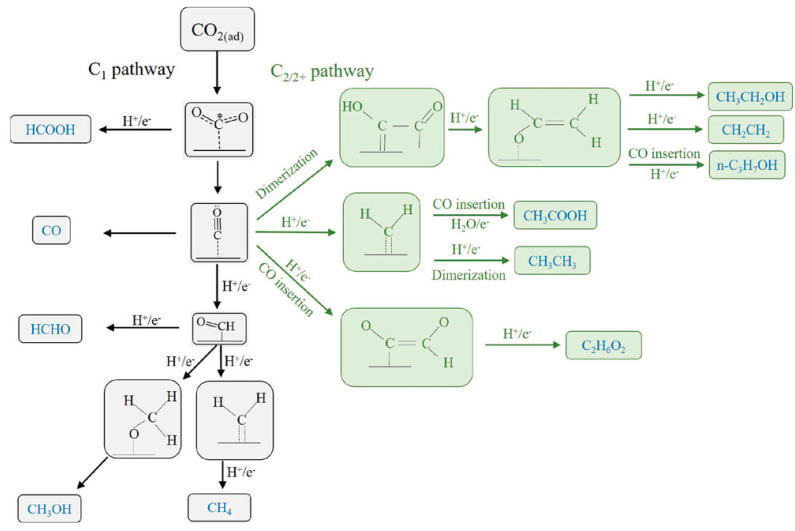
C_1_ and C_2/2+_ pathways on the Cu—based electrocatalyst. *OCO refers to the radical form of the adsorbed species. Reprinted with permission from Ref. [61]. Copyright 2021, Wiley-VCH GmbH.

**Figure 4 nanomaterials-13-01723-f004:**
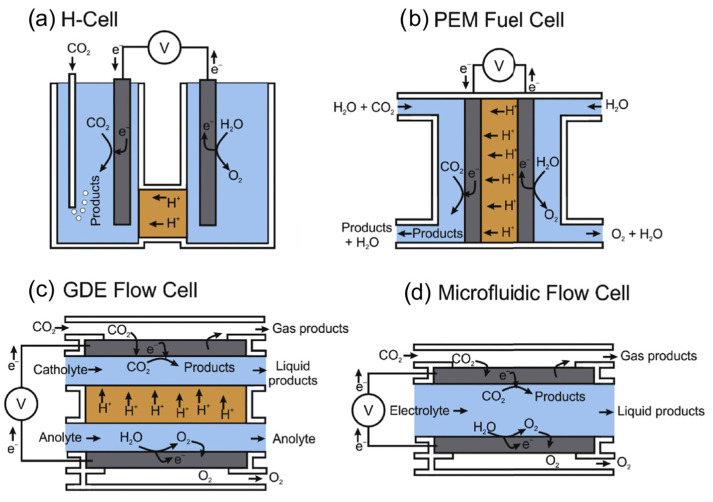
Representative examples of cell configurations employed for the CO_2_ER reaction. The main components are as follows: electrolytes (blue), catalyst or catalyst−loaded GDE (grey), ion exchange membrane (orange), and gas phase flow (white). (**a**) H−cell with the anode and cathode compartments separated by an ion−conductive membrane. A reference electrode can be placed in the cathodic compartment. (**b**) Proton exchange membrane (PEM) cell with a GDE in a membrane electrode assembly (MEA) set−up. CO_2_ typically enters the cathodic compartment dissolved in the electrolyte or as humidified gas. (**c**) GDE flow cell with the catholyte and anolyte separated via an ion exchange membrane. (**d**) Microfluidic flow cell with a single electrolyte flowing between the anode and cathode. Reproduced with permission from Ref. [83]. Copyright 2020, Elsevier B.V.

**Figure 5 nanomaterials-13-01723-f005:**
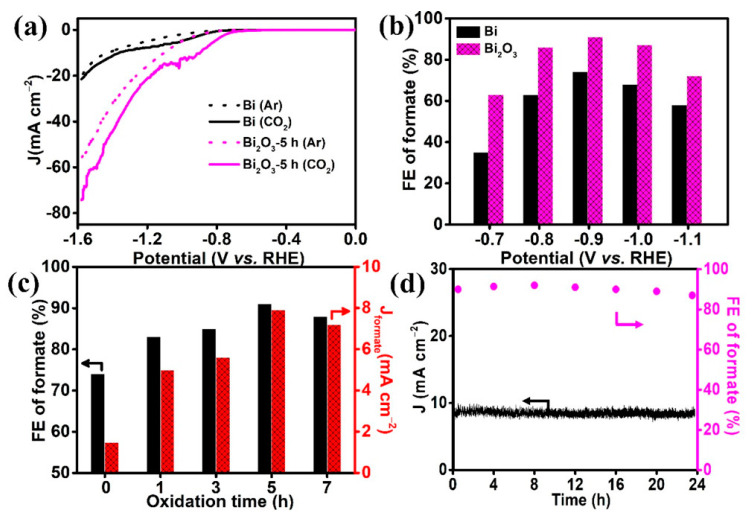
Bismuth−based catalysts for the CO_2_ electrochemical reduction to formate. (**a**) Linear sweep voltammograms of Bi and Bi_2_O_3_−5 h in Ar and CO_2_−saturated 0.5 M KHCO_3_ (scan rate: 10 mV s^−1^); (**b**) Faradaic efficiencies of formate for Bi and Bi_2_O_3_−5 h at different potentials; (**c**) Faradaic efficiencies and partial current densities of formate production over different bismuth oxides at −0.9 V vs. RHE; (**d**) stability test of Bi_2_O_3_−5 h at −0.9 V vs. RHE during 24 h. Reprinted with permission from Ref. [108]. Copyright 2020, American Chemical Society.

**Figure 6 nanomaterials-13-01723-f006:**
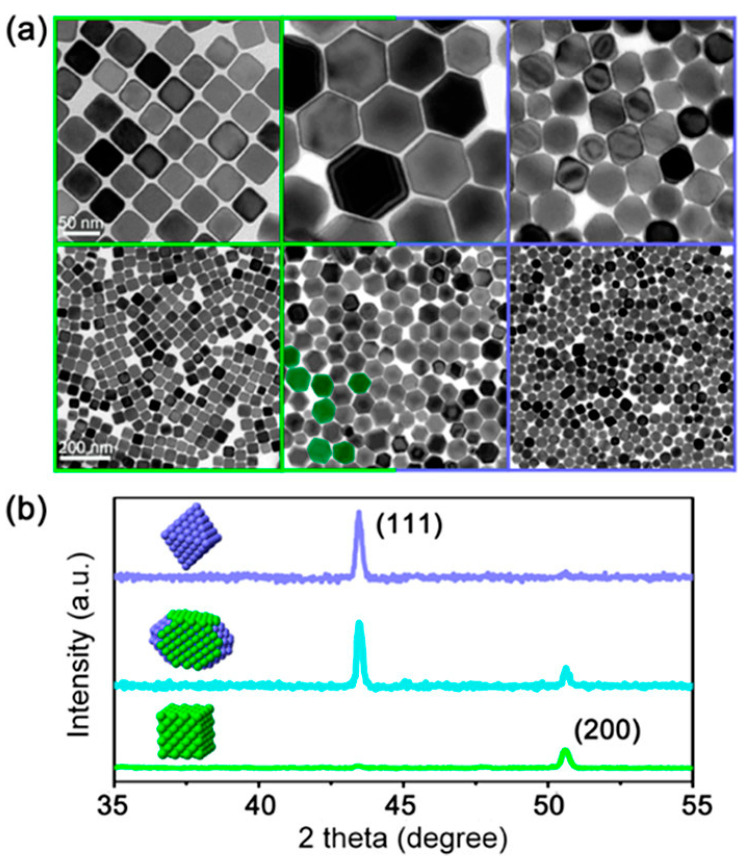
Cube-like, hexarhombic–docadehedron–like, and octahedron-like Cu single crystals. (**a**) (Top) TEM images of C–Cu (left), H–Cu (middle), and O–Cu (right), and (bottom) low-magnification images showing the overall condition with uniform size distribution. Some H–Cu nanoscale single crystals (bottom, middle) are green-coloured to emphasise the step surface features. (**b**) Powder X-ray diffraction (PXRD) pattern for C–Cu (green), H–Cu (cyan), and O–Cu (purple). Reprinted with permission from Ref. [134]. Copyright 2019, American Chemical Society.

**Figure 7 nanomaterials-13-01723-f007:**
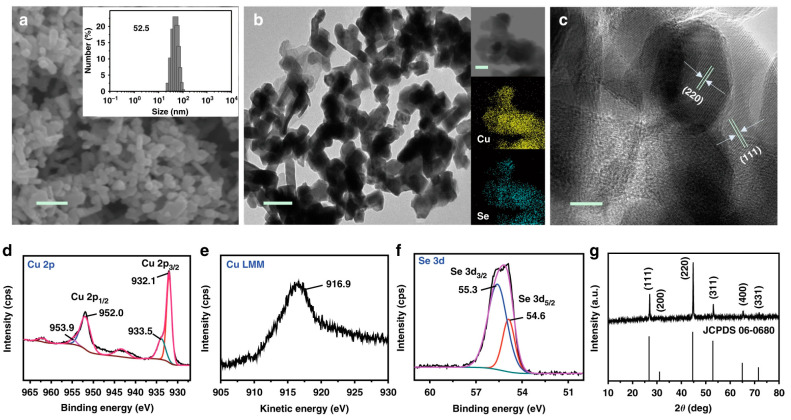
Characterisation of the copper selenide (Cu_1.63_Se(1/3)) nanocatalysts. (**a**) SEM image and (inset) size distribution of the nanocatalyst. Scale bar = 200 nm. (**b**) TEM image and (inset) corresponding elemental mappings. Scale bar = 100 nm. (**c**) HR-TEM image of the Cu_1.63_Se(1/3) nanocatalysts. Scale bar = 10 nm. XPS spectra: (**d**) Cu 2p, (**e**) Cu LMM, and (**f**) Se 3d. (**g**) XRD patterns of the Cu_1.63_Se(1/3) nanocatalysts. Reproduced with permission from Ref. [158]. Copyright 2019, Springer Nature.

**Table 1 nanomaterials-13-01723-t001:** Summary of the estimates of radiative forcing (with uncertainties) in 2011, relative to 1750. Data reported from 2013 IPCC report [3].

Drivers of Climate Change	Radiative Forcing Estimates (W m^−2^)
CO_2_	1.7 ± 0.4
CH_4_	1.0 ± 0.2
N_2_O	0.2 ± 0.1
Halocarbons (O_3_, CFCs, HCFCs)	0.2 ± 0.2
CO	0.2 ± 0.1
Solar irradiance	0.1 ± 0.1

**Table 2 nanomaterials-13-01723-t002:** Advantages and disadvantages of the most promising CO_2_ conversion routes.

CO_2_ Conversion	Advantages	Disadvantages	Ref.
Thermochemical	High productivity High selectivity	High temperature and pressure Thermal instability of the catalyst Sintering processes	[21,28,29]
Biochemical	Low temperature and pressure Non-toxic	Requires high costs Biomass harvesting and transportation Low productivity	[23,30,31]
Photochemical	No need for additional energy	Hard to scale up Low productivity Low selectivity	[32]
Electrochemical	Easy to scale up Renewable energies can be used Low temperature and pressure Easy tuning of the reaction outcomes	Request for high overpotential Scarce selectivity at high current densities	[33,34,35]

**Table 3 nanomaterials-13-01723-t003:** Faradaic efficiencies and current densities (J_tot_) of the main CO_2_ reduction products obtained using benchmark catalysts.

Product	Benchmark Catalyst	J_tot_ (mA cm^−2^)	FE (%)	Ref.
CO	5-fold twinned Ag NWs	~2.2	99.3	[151]
HCOOH	Bi-NRs@NCNTs	6.0	~91	[152]
CH_4_	SA-Zn/MNC	31.8	85	[157]
CH_3_OH	Cu_1.63_Se(1/3)	41.4	77.6	[158]
CH_3_CH_2_OH	Cu_n_ (n = 3 and 4) cluster	~2.0	91	[159]
n-propanol	CuS_X_—DSV	9.9	15.4	[160]
CH_3_COOH	CuMgAl LDH/CP	~0.3	84	[164]

## Data Availability

Data sharing is not applicable to this article.

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
