# Peer review of "From Traditional to New Benchmark Catalysts for CO2 Electroreduction"

_nanomaterials, 2023, doi:10.3390/nano13111723_

Round 1

Reviewer 1 Report

This review reports the most relevant examples of metal-based, nanostructured electrocatalysts for CO2ER proposed in the literature during the last years. Moreover, the benchmark materials are identified and the most promising strategies towards the selective conversion to high added value chemicals with superior productivities are highlighted. Overall, the review is professional and coherent. Therefore, it is recommended for publications in Nanomaterials after addressing the following minor comments.

1. In section 2, Page 4, line 131, what are the typical substances represented by C1, C2, and C3? Could the authors provide some examples?

2. In Page 6, line 181 when explaining “Moreover, the understanding of the reaction mechanisms that affect the electrocatalytic performances, alongside the changes in the catalyst structure, is of particular interest.” it would be better if there are specific examples to support it.

3. In Page 9, line 372 “Indeed, the thin oxide layers over the bulk electrode generally promote the formation of nanostructured interfaces, which will be discussed in the following paragraph”, studies achieved by means of advanced in situ characterisation techniques have reported the instability of the metal oxides species under the CO2 electroreduction conditions, however it seems that there is no the detailed discussion of in situ characterisation techniques.

4. In the sections of “Electrocatalysts of new Generation” and “Benchmark Electrocatalysts”, Cu-based catalysts were introduced both in the two sections, so the text can be concise in one part.

5. In the section of “Conclusions and future perspectives i”, the computational modelling was highly encouraged. However, the application of theoretical calculations in the CO2ER was not mentioned in the earlier part of the article, please provide some examples and explanations of the DFT in CO2ER.

6. Some reports related to CO2ER published recently (2022 and 2023) can be cited in the revision.

7. Illustrations facilitate readers' comprehension of the review article. More figures related the summary of electrolytic cell for electrochemical reaction and so on are encouraged.

Author Response

please see the file attached

Reviewer 2 Report

In this current review article by Serafini et.al. puts light on electrochemical CO2 reduction from traditional to recent benchmark catalysts. The article is encouraging, and well written. However, this articles lacks presenting recent benchmark articles on CO2 reductions. Many Bi-based electrocatalysts have been utilized as ec-CO2 reduction. E.g. ACS Catal. 2021, 11, 9, 4988–5003, Nano Lett. 2022, 22, 4, 1656–1664 etc. Authors should look into those articles and reference them. Moreover, as authors are discussing various materials/metals and their utilization in CO2 electroreduction, I found lack of literatures based on studies of crystal facet dependent electroreduction of CO2. Facet dependency plays a major role in developing new electrocatalysts. So, I would urge the author to include few benchmark examples in current review.

There are few minor issues which I think the authors should address before being considered for publication.

1.     As a review article, adding some figures as well as graphic abstract is highly encouraged.

2.     There are a few typos and grammatical mistakes in the manuscript, which needs to be rechecked and corrected.

3.     References are not formatted according to journal standard. It even shows invalid citation. Please check and correct.

Moderate rectification needed

Author Response

please see the file attached

Reviewer 3 Report

In this manuscript, “From Traditional to New Benchmark Catalysts for the CO2 Electroreduction” by Serafini et al. reviews the most relevant examples of metal-based, nanostructured electrocatalysts proposed in the literature during the last 40 years. This work is well collected the recent literature, but without adaption of some figures from literature, please see Advanced Energy Materials 9(24):1900090. Therefore, I would suggest that at least a revision may require before publication. Here are the comments and suggestions:

1.        The format of Table 2 could be revised.

2.        Authors are suggested to adapt some figures from literature with permission for easier understanding, please see Advanced Energy Materials 9(24):1900090.

Author Response

please see the file attached
